# IL-33-Induced Transcriptional Activation of LPIN1 Accelerates Breast Tumorigenesis

**DOI:** 10.3390/cancers13092174

**Published:** 2021-04-30

**Authors:** Jin-Young Kim, Garam Kim, Sung-Chul Lim, Hong-Seok Choi

**Affiliations:** 1College of Pharmacy, Chosun University, Gwangju 61452, Korea; kkk423@chosun.ac.kr (J.-Y.K.); garam@chosun.ac.kr (G.K.); 2Department of Pathology, School of Medicine, Chosun University, Gwangju 61452, Korea; clim@chosun.ac.kr

**Keywords:** IL-33, LPIN1, COT, JNK1/2, c-Jun

## Abstract

**Simple Summary:**

LPIN1 is a protein that exhibits dual functions as a phosphatidic acid phosphatase enzyme in the regulation of triglyceride and glycerophospholipid metabolism and a transcriptional coregulator. Recent advances in understanding suggest that LPIN1 frequently observed in various human cancer cell lines controls the main cellular processes in cancer progression. Therefore, there is considerable interest in the underlying mechanism regulating LPIN1 expression. We have determined for the first time that IL-33 regulates LPIN1 expression by recruiting c-Jun to the LPIN1 promoter. Consistent with these observations, IL-33 levels positively correlate with LPIN1 expression in human breast cancer. Our findings point to a critical role of IL-33-induced LPIN1 expression via COT/JNK1/2 pathway in promoting epithelial transformation and breast tumorigenesis.

**Abstract:**

Phospholipids are crucial materials that are not only required for cell membrane construction but also play significant roles as signaling molecules. LPIN1 is an enzyme that displays phosphatidate phosphatase activity in the triglyceride and phospholipid synthesis pathway. Recent studies have shown that overexpression of LPIN1 is involved in breast tumorigenesis, but the underlying mechanism regulating LPIN1 expression has not been elucidated yet. In the present study, we showed that the IL-33-induced COT-JNK1/2 signaling pathway regulates LPIN1 mRNA and protein expression by recruiting c-Jun to the LPIN1 promoter in breast cancer cells. IL-33 dose-dependently and time-dependently increased LPIN1 mRNA and protein expression. Moreover, IL-33 promoted colony formation and mammary tumorigenesis via induction of LPIN1 expression, while inhibition of LPIN1 disturbed IL-33-induced cell proliferation and mammary tumorigenesis. IL-33-driven LPIN1 expression was mediated by the COT-JNK1/2 signaling pathway, and inhibition of COT or JNK1/2 reduced LPIN1 expression. COT-JNK1/2-mediated IL-33 signaling activated c-Jun and promoted its binding to the promoter region of LPIN1 to induce LPIN1 expression. These findings demonstrated the regulatory mechanism of LPIN1 transcription by the IL-33-induced COT/JNK1/2 pathway for the first time, providing a potential mechanism underlying the upregulation of LPIN1 in cancer.

## 1. Introduction

Breast cancer is an abundant malignant tumor and the leading cause of cancer-related deaths among women worldwide [1]. The established risk factors for breast cancer are a woman’s age, family history of breast cancer, genetic configuration, pregnancies and reproductive treatment, consumption of alcohol, and exposure to ionizing radiation [2,3,4]. In addition, excess weight and obesity are regarded as promoting factors for breast cancer development and progression [5,6]. A meta-analysis suggested that obese patients had an approximately 30% increased risk of relapse or death as compared to women at a normal weight diagnosed with breast cancer [7,8]. Excess energy in the condition of obesity is stored as lipids, and the increased amount of lipids drastically alter the normal metabolic environment and create an environment that chronically transmits over-nutrition signals to cells [9].

Lipids are an essential component of the biological membranes and play a crucial role in energy storage and cell signaling. Several studies have shown that cancer cells undergo changes in various features of lipid metabolism, which provide the energy and molecules required for their rapid proliferation [10]. Alterations in lipid metabolism lead to changes in membrane composition, protein distribution, gene expression, and cellular function [11]. In particular, increased phospholipids markedly alter signal transduction, protect cancer cells from oxidative damage such as lipid peroxidation, and potentially inhibit the uptake of chemotherapeutic drugs [12]. As a metabolic enzyme for phosphatidic acid, LPIN1 converts phosphatidic acid to diacylglycerol during triacylglycerol and phospholipid biosynthesis [13,14]. Several studies have reported that LPIN1 is upregulated in several cancers, including breast cancer and prostate cancer, and its expression levels are positively correlated with the poor prognosis of triple-negative breast cancer and lung adenocarcinoma [15,16]. Knockdown of LPIN1 has been shown to reduce the proliferation and migration of breast and prostate cancer cells, suggesting that it functions as an oncogene in these cancers [17,18]. However, the underlying mechanism for the regulation of LPIN1 expression during transcription is poorly understood in breast tumorigenesis.

Interleukin-33 (IL-33), which is a member of the IL-1 family, has been shown to function in many essential physiological processes and be involved in the onset of many diseases, such as asthma, cardiovascular disease, obesity, and cancer [19]. IL-33 binds to suppression of tumorigenicity 2 receptor (ST2, also named interleukin 1 receptor-like 1) and interacts with interleukin-1 receptor accessory protein as a trimer complex to initiate downstream signaling. Events downstream of IL-33 stimulation include activation of p38 mitogen-activated protein kinases (p38-MAPK), c-Jun N-terminal kinase (JNK), and extracellular signal-regulated kinase (ERK), which activate downstream transcription factors, such as activator protein 1 (AP-1) [20,21,22,23]. IL-33/ST2 signaling exerts its effect by potently remodeling the tumor micro-environment (TME) by recruiting immune cells that secrete a diverse collection of molecules [24]. More recent studies have shown that IL-33 is a modulator that affects glucose uptake, glycolysis, and cellular insulin sensitivity by promoting cytokine release or interaction with targeted cells, resulting in changes in thermogenesis and glucose metabolism [25,26]. Furthermore, IL-33 plays a critical role in obesity-associated inflammation, atherosclerosis, and metabolic abnormalities, by promoting the production of T helper type 2 cytokines and polarizing macrophages toward a protective, alternatively-activated phenotype [27]. Although the function and mechanisms of IL-33 in various diseases have been identified, the regulation of LPIN1 expression in breast tumorigenesis has not been elucidated.

The aim of this study was to define the molecular mechanism of LPIN1 expression induced by IL-33 and to investigate its effect on breast tumorigenesis. We showed that IL-33 induced LPIN1 expression by regulating JNK/c-Jun signaling by activating the Cancer Osaka thyroid (COT). Inhibition of JNK1/2 or LPIN1 markedly inhibited IL-33-induced colony formation. Consistent with these observations, our results also showed that IL-33 induced mammary gland tumor development, whereas treatment with COT, JNK1/2, or LPIN1 inhibitor significantly attenuated tumor growth. These results highlight an IL-33 signaling pathway that conveys an oncogenic signal to promote aggressiveness in human breast cancer through the induction of LPIN1.

## 2. Materials and Methods

### 2.1. Cell Culture and Establishment of Stable Cell Lines

Dulbecco’s Modified Eagle’s Medium (DMEM), Roswell Park Memorial Institute (RPMI) medium, and fetal bovine serum (FBS) were purchased from Invitrogen (Carlsbad, CA, USA). HEK293 cells, obtained from the Korea Cell Line Bank (Seoul, Korea), were cultured in DMEM supplemented with 10% fetal FBS. 4T1 metastatic mouse breast carcinoma cells, a gift from Dr. Sang-Gun Ahn (Chosun University, College of Dentistry, Gwangju, Korea), were maintained in RPMI medium supplemented with 10% FBS. MCF7 cells, kindly provided by Dr. Kun Ping Lu (Beth Israel Deaconess Medical Center, Harvard Medical School, Boston, MA, USA), were cultured in DMEM supplemented with 10% FBS. All cell lines were cultured and maintained at 37 °C in a humidified atmosphere containing 5% CO_2_. HEK293 cells were used for protein-protein interaction analysis. MCF7 cells were used to study cell proliferation, colony formation, and a cell signaling pathway. 4T1 cells, which are part of an aggressive mouse mammary carcinoma cell line, were used to evaluate tumor growth. To establish IL-33 stably overexpressing cell lines, the human IL-33 expression plasmid, pCMV/Myc-IL-33, was transfected into mouse 4T1 cells. At 48 h after transfection, the cells were cultured in a medium containing 600 μg/mL neomycin. The antibiotic containing medium was replaced every 3–4 days. The resistant clones were pooled, transferred to a cell culture plate, and maintained with a selection medium. We designated the 4T1 cells transfected with the IL-33 expression vector as 4T1/Myc-IL-33 cells.

### 2.2. Antibodies and Reagents

The antibodies against LPIN1 (used at a dilution ratio of 1:1000), phosphorylated-COT (1:500), phosphorylated-JNK1/2 (1:500), and phosphorylated-c-Jun (1:1000) were purchased from Cell Signaling Technology (Danvers, MA, USA). Antibodies against MYC (1:5000) and *β*-actin (1:10,000) were from Sigma-Aldrich (St. Louis, MO, USA). ST2 (1:1000), COT (1:1000), JNK1 (1:1000), and JNK2 (1:1000) were purchased from Santa Cruz Biotechnology (Santa Cruz, CA, USA). Anti-V5 antibodies were obtained from Invitrogen (Carlsbad, CA, USA). Recombinant human IL-33 was obtained from R&D Systems (Minneapolis, MN, USA). The COT inhibitor TKI, PI3K/Akt inhibitor LY294002, STAT3 inhibitor WP1066, MEK inhibitor PD98059, and JNK inhibitor SP600125 were purchased from Calbiochem-Novabiochem (San Diego, CA, USA). The Ez-Chip kit and polyvinylidene difluoride (PVDF) membranes were purchased from Millipore (St. Louis, MO, USA). The ImProm-II™ Reverse Transcription System was purchased from Promega (Madison, WI, USA). The jetPEI^®^ cationic polymer transfection reagent was purchased from Polyplus-transfection (NY, USA). Lipofectamine™ LTX Reagent with PLUS™ transfection reagent was purchased from Invitrogen (Carlsbad, CA, USA).

### 2.3. Mammalian Expression Plasmids and Small Interfering RNA

The wild type pRK-Myc-COT plasmids were provided by Warner C Greene (University of California, San Francisco, CA, USA). The human IL-33 plasmid, pCMV/Myc-IL-33, and pcDNA3.1/V5-JNK1 were purchased from OriGene Technologies Inc. (Rockville, MD, USA). siRNA against JNK1 (accession number: NM_001323302.1) and JNK2 (accession number: NM_139068.2) were purchased from Mbiotech (Seoul, Korea). ST2 (accession number: NM_010743), COT (accession number: NM_005204), LPIN1 (accession number: NM_006221) SMARTpool^®^-specific and SMARTpool^®^-nonspecific control pool double-stranded RNA oligonucleotides were purchased from Dharmacon (Chicago, IL, USA). The siRNA sequence used for the JNK1/2 siRNA experiments was as follows: 5′-AAAAAGAAUGUCCUACCUUCU-3′.

### 2.4. Cell Proliferation Assay (Bromodeoxyuridine Incorporation)

The Cell Proliferation ELISA bromodeoxyuridine (BrdU) Kit used for this assay was purchased from Roche (Basel, Switzerland). Cells were seeded (0.5 × 10^5^ cells/well) and incubated at 37 °C in a 5% CO_2_-humidified atmosphere for 24 h. After 24 h, the cells were labeled with 10 µL/well BrdU labeling solution, and then re-incubated for an additional 4 h at 37 °C in a 5% CO_2_-humidified atmosphere. After removing the media, Fix Denat solution was added to each well, incubated at room temperature (RT) for 30 min, and then removed. An anti-BrdU-peroxidase working solution was added to each well and incubated for a further 90 min at RT. The cells were then washed with washing solution three times, and 100 µL of substrate solution was added to each well and incubated for 30 min. Cell proliferation was estimated by measuring absorbance at 370 nm.

### 2.5. RNA Extraction and Real-Time PCR

RNA was isolated from cells using the RNeasy^®^ Micro Kit (Qiagen, Chastsworth, CA, USA). cDNAs were prepared from the above RNAs using the ImProm-II™ Reverse Transcription System (Promega, Madison, WA, USA). Real-time PCR was performed using a StepOne™ Real-Time PCR System (Applied Biosystems, Foster City, CA, USA). The PCR mixture contained 10 ng of reverse-transcribed RNA, 10 nM of primers, and SYBR™ Green PCR Master Mix (Applied Biosystems, Foster City, CA, USA) in a final volume of 25 μL. The following primers were used for RT-PCR: human LPIN1 (sense, 5′-CCATTCACAGCGAGTCTTCA-3′, antisense, 5′-TGGAAGGGGAATCTGTCTTG-3′), human JNK1 (sense, 5′-CGTCTGGTGGAAGGAGAGAG-3′, antisense, 5′-TAATAACGGGGGTGGAGGAT-3′), and human JNK2 (sense, 5′-TCTGACGTCCTGGGCTGGAC-3′, antisense, 5′-GCAGCAGCCCTCAGGATCCT-3′).

### 2.6. Protein Immunoblotting and Immunoprecipitation

For immunoblotting, the cells were harvested and prepared using RIPA buffer containing 150 mM NaCl, 50 mM Tris-HCl (pH 7.4), 0.25% sodium deoxycholate, 1 mM EDTA, 1% NP40, 1 mM NaF, 0.2 mM PMSF, 0.1 mM sodium orthovanadate, and protease inhibitor cocktail (Roche, Switzerland). Protein samples were subjected to SDS-PAGE and transferred to a PVDF membrane, following which immunoblots were probed with the indicated antibodies. For immunoprecipitation, the cells were harvested and lysed in 150 mM NaCl, 50 mM Tris-HCl (pH 7.4), 0.25% sodium deoxycholate, 1 mM EDTA, 1% NP40, protease inhibitor cocktail (Roche, Switzerland), and phosphatase inhibitor cocktail. The lysates were incubated with rec-Protein G-Sepharose^®^ beads (Invitrogen, Carlsbad, CA, USA) and incubated overnight at 4 °C. The immunocomplexes were washed three times and recovered with 2× SDS sample buffer. The recovered immuno-complexes were subjected to SDS-PAGE and transferred to a PVDF membrane, following the immunoblots that were probed with the indicated antibodies. The immunoblots were visualized using chemiluminescence with the SuperSignal™ West Femto Maximum Sensitivity Substrate (Thermo Fisher Scientific, MA, USA) and detected using LAS4000-mini (Fujifilm, Tokyo, Japan). The whole western blot figures can be found in the Appendix A. 

### 2.7. Chromatin Immunoprecipitation

ChIP analysis was performed using an EZ-Chip Kit. For crosslinking, the cells were treated with 18.5% formaldehyde for 10 min. Then, the reaction was stopped by adding 10 × glycine for 5 min. The cells were washed twice with a cold phosphate-buffered saline (PBS)-containing protease cocktail. Following that, the cells were lysed in SDS lysis buffer and sonicated three times for 30 s at intervals of 10 s. Chromatin solutions were precipitated using c-Jun antibody and incubated overnight at 4 °C. Protein G agarose was added to recover the immune complex for 1 h 30 min at 4 °C. To reverse the protein/DNA, crosslinks were treated with 5 M NaCl and incubated overnight at 65 °C. Next, the cells were treated with RNase A for 30 min at 37 °C, then treated with proteinase K for 2 h at 45 °C, and purified using a spin column. Eluted DNA was analyzed using RT-PCR, and the product was loaded onto a 2% agarose gel for electrophoresis and visualized under UV illumination post SYBR™ Gold Nucleic Acid staining (Invitrogen, Carlsbad, CA, USA). The ChIP-enriched DNA was subjected to PCR using the following LPIN1 primers: sense, 5′-GCTCTTCCGAGTGCAGCTAGG-3′ and antisense, 5′-GCAAATTAGGTCCCGCTGCTG-3′.

### 2.8. Anchorage-Independent Cellular Transformation Assay (Soft Agar Assay)

Wells of a six-well plate were coated with 3 mL of 0.8% soft agar base agar with SP600125 or propranolol in the presence or absence of IL-33. After the agar has solidified, 8 × 10^3^ cells were exposed to SP600125 or propranolol, in the presence or absence of IL-33 in 1ml of a 0.3% soft agar layer containing basal medium Eagle agar containing 10% FBS, 2 mM l-glutamine, and 25 μg/mL gentamicin. The cultures were maintained at 37 °C in a 5% CO_2_ incubator. After 14 days, cell colonies were scored using an Axiovert 200M fluorescence microscope and Axio Vision software (Carl Zeiss, Thornwood, NY, USA).

### 2.9. Tumorigenicity Assay in BALB/c Mice

Six-week-old female BALB/c mice (18–20 g) were obtained from Samtako Co. (Osan, Korea), acclimatized for one week, and maintained in a clean room at the College of Pharmacy, Chosun University (Gwangju, Korea). The protocols for the animal studies were approved by the Animal Care Committee of Chosun University. Mice were randomly divided into five groups of 20 animals each. Stably transformed 4T1 mouse breast cancer cell lines (mock or Myc-IL-33) were suspended in PBS, and the cell suspensions were treated with TKI (100 μM), SP600125 (100 μM), or propranolol (500 mM) that were injected with the mammary gland of the mice at a concentration of 1 × 10^6^ cells/100 μL. Mice were euthanized by cervical dislocation at 14 days after the first injection and tumor volume and weight were calculated in all groups. The tumor volume was calculated using the formula: V = (ab^2^)/2, where ‘a’ was the longest diameter and ‘b’ was the shortest diameter of the tumor.

### 2.10. Tumor Samples

Informed consent was obtained from all the patients, and research protocols were approved by the ethics committee of Chosun University Hospital. The breast tissues that were selected for immunohistochemical staining were collected from a group of 60 patients with breast cancer (age range: 42–72 years).

### 2.11. Immunohistochemical Staining Analysis

The expression level of IL-33 and LPIN1 in breast cancer tissue were determined by IHC. Immunolocalization for each antibody was performed using a Polink-2 HRP Plus mouse diaminobenzidine detection system (Golden Bridge International, Mukilteo, WA, USA), according to the supplier’s protocol. Slides were incubated for 1 h with anti-IL-33 and anti-LPIN1 antibodies in a moist chamber at 37 °C. Distinct cytoplasmic staining for IL-33 and LPIN1 were considered to indicate positive immune reactivity.

### 2.12. Quantification and Statistical Analysis

Data from the real-time PCR, BrdU incorporation, and soft agar assays were analyzed using unpaired *t*-tests, and *p*-values less than or equal to 0.05 were considered statistically significant. Statistical calculations were performed using PRISM software for Macintosh version 5.0 (GraphPad, La Jolla, CA, USA).

## 3. Results

### 3.1. IL-33 Induces LPIN1 mRNA and Protein Expression

It is reported that the expression of LPIN1 is regulated by lipopolysaccharides (LPS) and cytokines, which are involved in inflammation. To examine whether IL-33 affects LPIN1 expression, we measured the mRNA and protein levels of LPIN1 in MCF7 cells exposed to IL-33. The mRNA levels of LPIN1, as determined using real-time PCR (qRT-PCR) or reverse transcription PCR (RT-PCR), were markedly increased upon treatment with IL-33 (Figure 1A,B).

Moreover, IL-33 induced the protein expression of LPIN1 in a dose-dependent and time-dependent manner (Figure 1C,D). Treatment with exogenous IL-33 induced LPIN1 expression, which may enhance LPIN1 expression through the autocrine activity of endogenously expressed IL-33. To investigate this mechanism, we transfected Myc-IL-33 into MCF7 cells. The results showed that overexpression of IL-33 led to upregulation of LPIN1 (Figure 1E). IL-33 is a ligand of the receptor ST2. To determine whether ST2 is involved in IL-33-induced LPIN1 expression, MCF7 cells were exposed to IL-33 post transfection with small interfering RNA (siRNA)-ST2. Knockdown of ST2 markedly suppressed the induction of LPIN1 by IL-33 (Figure 1F). Together, these data suggest that the IL-33/ST2 axis upregulates the mRNA and protein expression of LPIN1 in MCF7 cells.

### 3.2. COT Mediates LPIN1 Expression Induced by IL-33

A previous study showed that COT acts as a mediator of the IL-33/ST2 signaling pathway. To determine the effect of COT on LPIN1 expression, we overexpressed COT in MCF7 cells. The results showed that there was a gradual increase in the LPIN1 protein levels upon overexpression of COT, indicating that COT mediates endogenous LPIN1 expression (Figure 2A).

We further examined the effect of COT on IL-33-induced LPIN1 expression. IL-33-induced LPIN1 expression was enhanced upon COT overexpression (Figure 2B), while it was reduced upon silencing COT in MCF7 cells (Figure 2C). Next, to explore the effects of COT inhibition on IL-33-induced LPIN1 expression, MCF7 cells were pretreated with a specific inhibitor of COT, TKI (Tpl2 kinase inhibitor; 4-(3-chlor-4-fluorophenylamino)-6-(pyridin-3-yl-methylamino)-3-cyano-1,7-naphthylridine). Real-time PCR and RT-PCR analyses revealed that the IL-33-mediated increase in LPIN1 mRNA levels was attenuated by TKI in MCF7 cells (Figure 2D,E). Furthermore, TKI treatment decreased IL-33-induced LPIN1 protein levels (Figure 2F). Together, these results suggested that IL-33-induced LPIN1 expression is regulated by COT in MCF7 cells.

### 3.3. JNK1/2 Plays a Key Role in IL-33/ST2/COT-Mediated LPIN1 Expression

COT has been reported to mediate the MEK-ERK, JNK1/2, phosphatidylinositol-3-kinase (PI3K)/Protein kinase B (Akt), Jak/signal transducer, and activator of transcription (STAT) pathways induced by IL-33 [23] Given the important role of the IL-33/ST2/COT axis in the regulation of LPIN1 expression, we examined which COT signaling cascades are involved in IL-33-induced LPIN1 expression. IL-33 was treated with specific inhibitors of MEK-ERK, JNK1/2, PI3K/Akt, and Jak/STAT pathways, such as PD98059 (MEK inhibitor), SP600125 (JNK inhibitor), LY294002 (PI3K/Akt inhibitor), and WP1066 (STAT3 inhibitor) in MCF7 cells. The results showed that the IL-33-induced LPIN1 mRNA and protein levels were strongly reduced by JNK1/2 inhibitors (Figure 3A).

In contrast, PD98059, LY294002, and WP1066 failed to block IL-33-induced LPIN1 mRNA and protein expression (Figure 3A). Consistent with these results, IL-33-mediated LPIN1 expression was blocked by SP600125 in a dose-dependent manner (Figure 3B). To further determine whether JNK1/2 affects IL-33-induced LPIN1 levels mediated by COT, MCF7 cells were transfected with MYC-COT and exposed to IL-33 with or without a JNK1/2 inhibitor. IL-33-induced LPIN1 expression was enhanced upon overexpression of COT, but SP600125 treatment strongly reduced LPIN1 expression to the same extent as that displayed in mock-transfected control MCF-7 cells (Figure 3C). These results indicated that IL-33-induced LPIN1 expression is COT-JNK pathway-dependent. Since the COT-JNK1/2 axis is important for regulating IL-33-induced LPIN1 expression, we sought to determine whether COT and JNK interact physically. An in vitro interaction study was conducted by co-transfecting HEK293 cells with plasmids expressing either MYC-COT, V5-JNK1, or -JNK2. Immunoprecipitation/immunoblotting revealed COT bound to JNK1 and JNK2 (Figure 3D). To further examine whether IL-33 affected the endogenous interaction between COT and JNK1 or JNK2, we performed immunoprecipitation following IL-33 treatment in MCF7 cells. The results showed that IL-33 enhanced the endogenous interaction between COT and JNK1 or JNK2 in a dose-dependent manner (Figure 3E,F). Next, to examine whether JNK1 and JNK2 act as mediators of IL-33-induced LPIN1 expression, we first assessed the effect of JNK1 or JNK2 overexpression on LPIN1 levels. As expected, MCF-7 cells transfected with different amounts of V5-JNK1 or V5-JNK2 gradually increased the expression of LPIN1 (Figure 3G). In addition, the expression of LPIN1 in IL-33 stimulation was enhanced upon overexpression of JNK1 or JNK2 (Figure 3H), whereas JNK1 and JNK2 knockdown effectively blocked LPIN1 expression upon IL-33 stimulation (Figure 3I). We observed that COT-JNK is a critical signal transducer for the regulation of LPIN1 expression by IL-33 induction in breast cancer cells.

### 3.4. JNK1/2 Induces LPIN1 Expression through Increased Association of c-Jun with the LPIN1 Promoter

Given that the IL-33/ST2/COT/JNK1/2 cascade regulates the levels of LPIN1, we examined whether JNK1 or JNK2 regulates the transcription levels of LPIN1. MCF7 cells were transfected with different amounts of V5-JNK1 or V5-JNK2. The results showed that *LPIN1* mRNA levels gradually increased upon overexpression of JNK1 or JNK2 in MCF-7 cells (Figure 4A,B).

Based on our observations of increased *LPIN1* mRNA levels following IL-33 treatment, we next examined the effect of JNK1/2 on IL-33-induced *LPIN1* mRNA levels. MCF7 cells were transfected with siRNA-control or siRNA-JNK1/2 and treated with IL-33. The results showed that the increase in LPIN1 mRNA levels mediated by IL-33 were reduced upon JNK1/2 knockdown in MCF7 cells (Figure 4C). Similarly, IL-33-mediated LPIN1 mRNA levels were reduced in a dose-dependent manner by SP600125 (Figure 4D). As shown in Figure 3, IL-33 led to the activation of JNK and its downstream molecule c-Jun. The transcriptional activity of c-Jun is regulated by the phosphorylation of SAPK/JNK. Given that the transcriptional level of LPIN is regulated by the IL-33-COT-JNK pathway, to determine whether activated c-Jun regulates the LPIN1 promoter, we analyzed the LPIN1 promoter for regulatory DNA binding elements and identified a c-Jun binding site located within the LPIN1 promoter, based on *Homo sapiens* chromosome 2, GRCh38 (Figure 4E). To investigate whether c-Jun could bind to the *LPIN1* promoter, we performed a chromatin immunoprecipitation (ChIP) assay in MCF7 cells. The results of the ChIP assay showed that endogenous c-Jun bound to the -2414 to -2112 region of the *LPIN1* promoter (Figure 4F), indicating that LPIN1 transcription levels can be directly regulated by c-Jun. Upon IL-33 stimulation, activated JNK1/2 signaling has been suggested to play a crucial role in LPIN1 transcriptional activation. Therefore, we investigated whether IL-33 treatment induces the binding of c-Jun to the LPIN1 promoter. Binding of c-Jun to the *LPIN1* promoter was enhanced upon IL-33 stimulation. Inhibition of JNK by SP10062, however, remarkably blocked IL-33-enhanced c-Jun interaction with the promoter (Figure 4G). Taken together, these findings suggest that IL-33 induces LPIN1 mRNA via c-Jun association with the *LPIN1* promoter.

### 3.5. LPIN1 Overexpression Enhances Breast Cancer Cell Transformation Induced by IL-33

It has been reported that IL-33 is associated with tumorigenesis, metastasis, and proliferation of tumor cells. To investigate whether the COT/JNK pathway is essential for IL-33-induced tumorigenesis, we first examined the effect of SP600125 on IL-33-induced cell proliferation and anchorage-independent cell transformation in MCF7 cells. The results showed that SP600125 treatment inhibited IL-33-induced proliferation of MCF7 cells (Figure 5A).

In the soft agar colony-forming assay, we observed that IL-33 increased the colony numbers of MCF-7 cells. The same was found to be repressed in the presence of various doses of SP600125 (Figure 5B,C). Consistent with this, the LPIN1 inhibitor propranolol suppressed IL-33-induced proliferation of MCF-7 cells in a dose-dependent manner (Figure 5D). Treatment with propranolol also reduced an IL-33-mediated increase in the number of MCF-7 cell colonies in the soft agar assay (Figure 5E,F), indicating that LPIN1 induced by the COT/JNK1/2 pathway is essential for anchorage-independent proliferation of breast cancer under IL-33 stimulation.

### 3.6. LPIN1 Abundance Is Positively Correlated with IL-33 Expression and Promotes Breast Mammary Tumorigenesis

To evaluate the effects of IL-33-induced LPIN1 on breast tumorigenesis, 4T1 cells stably transfected with Myc-IL-33 were established (4T1/Myc-IL-33, Figure 6A) and subsequently injected into the mammary gland of BALB/c mice, along with TKI, SP600125, or propranolol. Representative tumor images demonstrated that 4T1-IL-33 tumors markedly increased both the volume and weight of the tumors, and that these effects were efficiently restrained by the inhibitors (Figure 6B,C).

These findings suggest that IL-33/COT/JNK1/2-induced LPIN1 plays a significant role in breast tumorigenesis. To understand the pathological relevance of IL-33 and LPIN1 in breast tumorigenesis, immunohistochemistry was performed on human breast cancer tissues using anti-IL-33 and anti-LPIN1 antibodies. Among eleven breast cancer samples with low IL-33, ten samples also showed lower LPIN1 expression. In contrast, five of the nine IL-33-high samples also had higher expression of LPIN1, indicating a positive correlation of IL-33 with LPIN1 levels in human breast cancer (Figure 6D,E). Similarly, the Cancer Genome Atlas (TCGA) data showed that IL-33 and LPIN1 expression levels were positively correlated in the breast cancer patients (Figure 6F).

## 4. Discussion

Cancer cells reprogram their metabolism to increase the synthesis of macromolecules for rapid proliferation. It is known that de novo fatty acid synthesis occurs at a very high rate in tumor tissues [28]. Compared to fatty acids, much less is known about the synthesis of phospholipids, which are essential for membrane biogenesis in cancer cells [29]. LPIN, a phosphatidic acid phosphatase (PAP), controls the rate-limiting step in the phospholipid synthesis pathway. The three LPIN1 proteins (LPIN1, LPIN2, and LPIN3) each have PAP activity, but LPIN1 is a predominant PAP enzyme in many metabolic tissues [30]. In humans, LPIN1 is highly expressed in muscle and adipose tissue, which is consistent with its role in lipid metabolism in these tissues [31,32]. The high expression of LPIN1 has been shown in various cancers, including breast cancer and prostate cancer, and its overexpression correlates with poor prognosis in these patients [16]. In addition, LPIN1 depletion has been shown to strongly inhibit the proliferation of cancer cells, with only a limited effect on non-cancerous cells [17]. Regulated LIPIN1 expression plays a crucial role in many biological processes, including cell proliferation and differentiation. However, the signaling pathway that mediates the induction of LPIN1 has not yet been studied. In this study, we demonstrated for the first time that the IL-33/ST2 signaling pathway plays an important role in regulating LPIN1 expression, by enhancing the association of the c-Jun transcription factor with the LPIN1 promoter in breast cancer cells. Inhibition of the JNK1/2 signaling pathway failed to increase IL-33-stimulated LPIN1 expression. Furthermore, propranolol and SP600125, which are LPIN1 and JNK1/2 inhibitors, respectively, strongly suppressed IL-33-induced colony formation in breast cancer cells and mammary gland tumor development.

IL-33 is a more recently identified IL-1 family member and a ligand for ST2 [33]. IL-33/ST2 signaling has been reported to promote breast cancer in several studies. In addition, IL-33 and ST2 levels have been found to be higher in breast tissues with cancer, as compared to healthy breast tissues [23], which may suggest a role for the IL-33/ST2 axis in tumor progression. IL-33 binds to a heterodimeric receptor complex consisting of an ST2 and IL-1R accessory protein and induces the activation of numerous signaling proteins, including NF-κB, inhibitor of NF-κB-*α* (IκB-*α*), ERK1/2, p38, JNK1/2, and COT [34]. The data presented in this study showed that the IL-33/ST2 pathway strongly increases LPIN1 expression. In addition, ST2 knockdown lowered IL-33-induced LPIN1 expression, indicating that LPIN1 expression depends on the IL-33 signaling pathway via ST2. We also demonstrated that overexpression of COT enhanced the expression of LPIN1 induced by IL-33. In contrast, silencing of COT and treatment with TKI, which is a COT inhibitor, strongly suppressed IL-33-induced LPIN1 expression. Overall, these results indicated that COT is an important mediator of the IL-33/ST2 signaling pathway for LPIN1 expression. To further understand the pathophysiological relevance of IL-33-induced LPIN1 expression in breast tumorigenesis, an immunohistochemistry assay, and a TCGA database were analyzed. We found that IL-33 levels were positively correlated with LPIN1 levels in breast cancer tissues, indicating that the IL-33 facilitates breast carcinogenesis via enhanced LPIN1 expression. Consistent with these findings, LPIN1 is significantly up-regulated in basal like triple-negative breast cancer and overexpression of LPIN1 correlates with poor patient survival [16]. These reports support our work that high levels of LPIN1 play an important role in tumor progression in breast cancer.

COT is a member of the MAP3K serine/threonine protein kinase family, which plays an important role in inflammation, immunity, and oncogenic events. Moreover, COT is closely associated with cytokine release from inflammatory cells, which has crucial effects on both tumor cells and the TME [35]. The TME is a central operating system for the progression of tumor development, local invasion, and metastasis [36]. COT activates several signaling pathways such as MAPK, NF-κB, AP-1, and STAT3 in response to various pro-inflammatory stimuli including bacterial LPS, tumor necrosis factor-*α*, cluster of differentiation 40, IL-1β, IL-22, and IL-33, to induce innate immune responses and tumorigenesis [23,37,38,39]. Previous studies have reported that COT mediates the MEK1/2, JNK1/2, STAT, and AKT signaling pathways through IL-33, leading to breast tumorigenesis [23]. We found that, among the different COT-mediated IL-33 signaling pathways, the JNK1/2 pathway regulates LPIN1 expression. JNK1/2 overexpression strongly induced LPIN1 expression, whereas treatment with the inhibitor of JNK1/2 attenuated IL-33-induced LPIN1 expression. Moreover, there was an increase in IL-33-induced LPIN1 expression upon overexpression of COT, but the same was strongly reduced upon JNK inhibition. COT has been reported to contribute to the optimal activation of JNK1/2, specifically in response to genotoxic stress or inflammatory stimuli, such as immune cytokines [39,40]. However, little is known about the direct interaction and understanding of COT and JNK1/2 in cytokine responses. In the present study, we showed that COT directly interacts with JNK1/2, and this endogenous interaction is enhanced by IL-33. These results show that COT and JNK1/2 are critical components of the signal transduction pathways that link the engagement of LPIN1 expression by IL-33 induction in breast cancer cells.

Moreover, COT-JNK1/2 increased IL-33-induced phosphorylation of c-Jun at ser63, indicating that ser63 phosphorylation of c-Jun enhanced the transcriptional activity of c-Jun target genes. The phosphorylation of c-Jun by JNK can form heterodimers or homodimers with other factors and induce the transcription of many genes [41]. One transcriptional target of c-Jun is the CCND1 gene, which encodes the G1 to the S phase regulator cyclin D1 [42]. Moreover, c-Jun induces the abundance of the stem cell factor and chemokine ligand 5 through transcriptional activation of gene promoters, promoting the self-renewal and tumor invasiveness of mammary tumor stem cells [43]. Several studies have suggested that c-Jun is also involved in apolipoprotein gene expression and lipid lipoprotein homeostasis, and it particularly suppresses the expression of the key adipogenic transcription factors CCAAT/enhancer-binding protein *α* and PPAR*γ* [44]. The regulation of c-Jun is known to be an important event in various diseases, including cancer [43,45,46], but the influence of c-Jun on LPIN1 overexpression and its significance in oncogenesis remain unknown. In this study, we found that c-Jun is able to bind to the LPIN1 promoter, to upregulate LPIN1 expression. The LPIN1 proximal promoter harbors the c-Jun binding site, which consists of the nucleotide sequence 5′-TGGCTGAGTCACTG-3′, and the results of the ChIP assay showed that endogenous c-Jun binds to the *LPIN1* promoter. In addition, there was an increase in the binding of c-Jun to the LPIN1 promoter upon IL-33 stimulation. In contrast, the IL-33-induced c-Jun association of the LPIN1 promoter was reduced upon JNK inhibition. These results indicated that the IL-33-mediated expression of LPIN1 is regulated by c-Jun binding to the LPIN1 promoter.

## 5. Conclusions

In the present study, we demonstrated that the IL-33/ST2/COT/JNK1/2 signaling pathway could overexpress LPIN1 in breast cancer cells. Inhibition of LPIN1 upon propranolol treatment suppressed IL-33-induced colony formation and mammary tumorigenesis. Consistent with these observations, IL-33 levels were positively correlated with LPIN1 expression in human breast cancer cells. More importantly, we found that COT-JNK1/2 is a pivotal signaling transducer for the regulation of LPIN1 expression in breast cancer cells. Inhibition of JNK1/2 diminished IL-33-induced LPIN1 expression and decreased cell proliferation and colony formation in MCF7 cells. Our data clearly show that IL-33 is a potent inducer of LPIN1 expression via the COT-JNK1/2 pathway and may play a significant role in breast tumorigenesis.

## Figures and Tables

**Figure 1 cancers-13-02174-f001:**
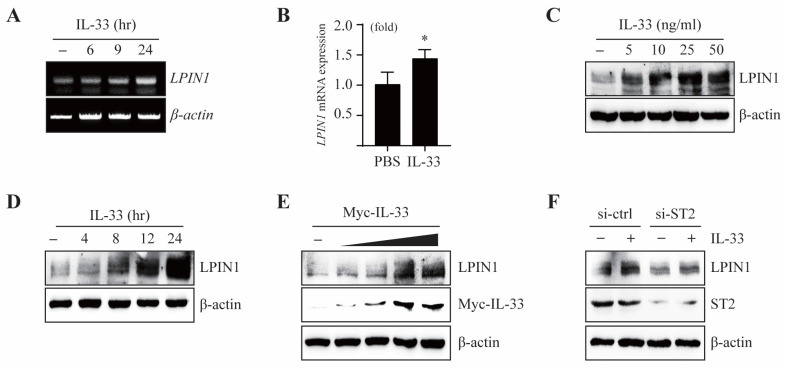
IL-33 regulates transcriptional activation of LPIN1. (**A**,**B**) MCF7 cells were serum starved for 24 h and then treated with IL-33 for the indicated time periods. The levels of LPIN1 and *β*-actin mRNA were determined using RT-PCR and real-time PCR. Columns, mean of triplicate samples; bars, S.D. *, *p* < 0.05, when compared to the control cells. (**C**,**D**) MCF7 cells were serum starved for 24 h, treated with the indicated doses of IL-33 for 24 h (**C**) or with 25 ng/mL IL-33 for the indicated times (**D**), harvested, and lysed. Proteins in the whole cell lysates were separated using SDS-PAGE and subjected to immunoblotting. (**E**) MCF7 cells were transfected with different amounts of pcDNA4/Myc-IL-33, incubated for 48 h, harvested, and subjected to immunoblotting. (**F**) MCF7 cells were transfected with siRNA-control or siRNA-ST2. At 48 h after transfection, the cells were serum starved for 24 h, treated with 25 ng/mL IL-33 for 24 h, harvested, and lysed. Proteins in the whole cell lysates were separated using SDS-PAGE and subjected to immunoblotting.

**Figure 2 cancers-13-02174-f002:**
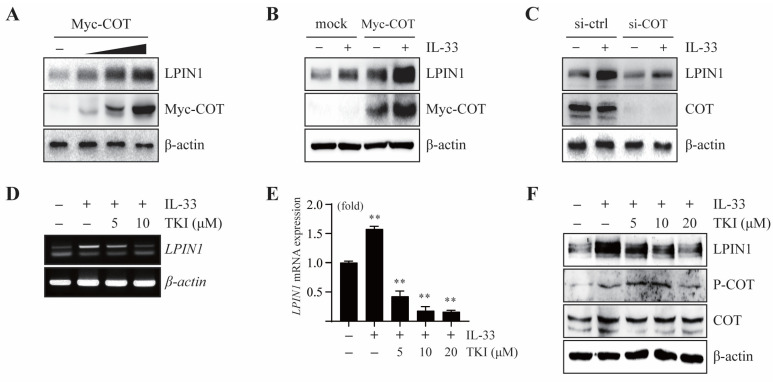
COT mediates IL-33-induced LPIN1 expression. (**A**) MCF7 cells were transfected with different amounts of Myc-COT, incubated for 48 h, harvested, and subjected to immunoblotting. (**B**,**C**) MCF7 cells were transfected with Myc-COT (**B**) or siRNA-COT (**C**). At 48 h after transfection, the cells were serum starved for 24 h, then treated with 25 ng/mL IL-33 for 24 h, harvested, and lysed. Proteins in the whole cell lysates were separated using SDS-PAGE and subjected to immunoblotting. (**D**,**E**) MCF7 cells were serum starved for 24 h, pretreated with the indicated concentrations of TKI for 2 h, exposed to 25 ng/mL IL-33 for 24 h, and harvested. The mRNA levels were assessed using RT-PCR and real-time PCR. Columns, mean of triplicate samples. bars, S.D. **, *p* < 0.01, when compared to the control cells. (**F**) MCF7 cells were serum starved for 24 h, pretreated with the indicated concentrations of TKI for 2 h, then exposed to 25 ng/mL IL-33 for 24 h, harvested, and lysed. Proteins in the whole cell lysates were separated using SDS-PAGE and subjected to immunoblotting.

**Figure 3 cancers-13-02174-f003:**
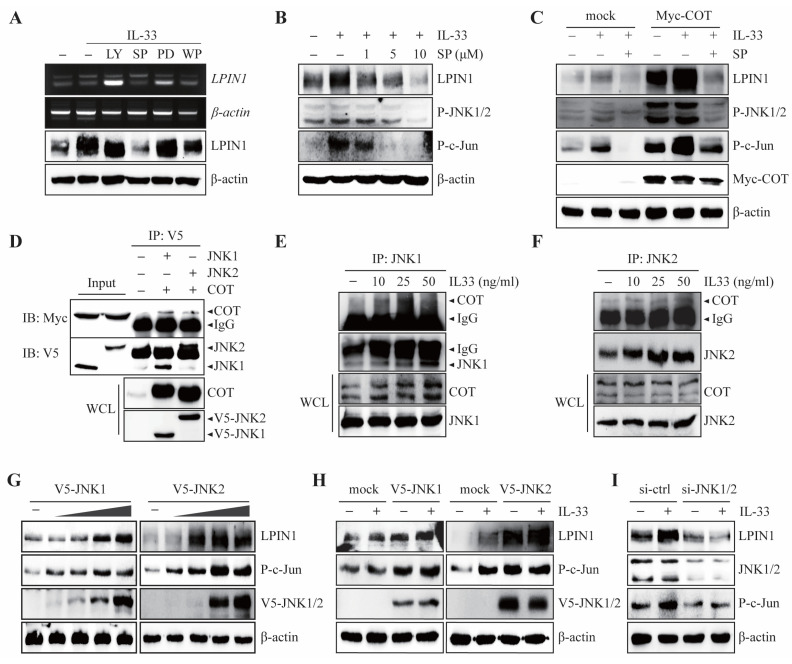
COT interacts with JNK1/2 to mediate IL-33-induced LPIN1 expression. (**A**) MCF7 cells were serum starved for 24 h, pretreated with the indicated inhibitors for 2 h, then exposed to 25 ng/mL IL-33 for 24 h, and harvested. The protein and mRNA levels were assessed using immunoblotting or RT-PCR. LY294002 (PI3K inhibitor). SP600125 (JNK inhibitor). PD98059 (ERK inhibitor). WP1066 (STAT3 inhibitor). (**B**) MCF7 cells were serum starved for 24 h, pretreated with the indicated concentrations of SP600125 for 2 h, then exposed to 25 ng/mL IL-33 for 24 h, harvested, and lysed. Proteins in the whole cell lysates were separated using SDS-PAGE and subjected to immunoblotting. (**C**) MCF7 cells were transfected with MOCK or Myc-COT. At 48 h after transfection, the cells were serum starved for 24 h, pretreated with the indicated concentrations of SP600125 for 2 h, then exposed to 25 ng/mL IL-33 for 24 h, harvested, and lysed. Proteins in the entire cell lysates were separated using SDS-PAGE and subjected to immunoblotting. (**D**) HEK293 cells expressing V5-JNK1, V5-JNK2 with Myc-COT were subjected to immunoprecipitation with anti-V5 antibodies, which is followed by immunoblotting with anti-Myc or anti-V5 antibodies. (**E**,**F**) MCF7 cells were starved for 24 h, exposed to 25 ng/mL IL-33 for the indicated time, harvested, and lysed. Immunoprecipitation was performed using JNK1 (**E**) or JNK2 (**F**) antibodies and then analyzed using immunoblotting, as indicated. (**G**) MCF7 cells were transfected with different amounts of pcDNA4/V5-JNK1 (left) or pcDNA4/V5-JNK2 (right), incubated for 48 h, harvested, and subjected to immunoblotting. (**H**) MCF7 cells were transfected with either V5-JNK1 (left) or V5-JNK2 (right). At 48 h after transfection, the cells were serum starved for 24 h, treated with 25 ng/mL IL-33 for 24 h, harvested, and lysed. Proteins in the whole cell lysates were separated using SDS-PAGE and subjected to immunoblotting. (**I**) MCF7 cells were transfected with siRNA-JNK1/2. At 48 h after transfection, the cells were serum starved for 24 h, treated with 25 ng/mL IL-33 for 24 h, harvested, and lysed. Proteins in the whole cell lysates were separated using SDS-PAGE and subjected to immunoblotting.

**Figure 4 cancers-13-02174-f004:**
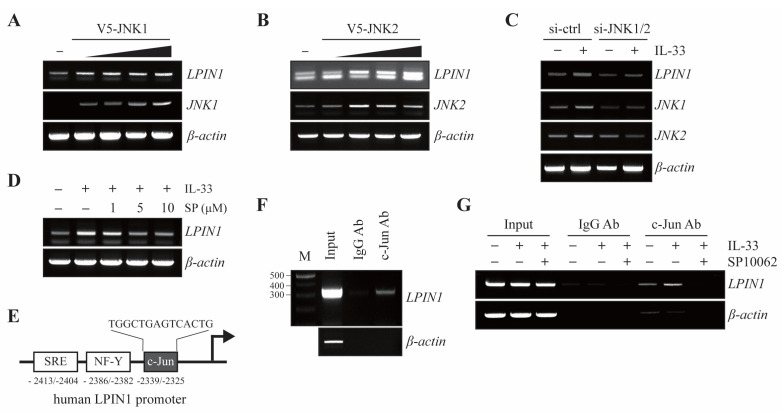
IL-33 stimulates binding of c-Jun to the LPIN1 promoter. (**A**,**B**) MCF7 cells were transfected with different amounts of pcDNA4/V5-JNK1 (**A**) or pcDNA4/V5-JNK2 (**B**), incubated for 48 h, harvested, and subjected to immunoblotting. The levels of LPIN1 and *β*-actin mRNA were determined using RT-PCR. (**C**) MCF7 cells were transfected with siRNA-JNK1/2. At 48 h after transfection, the cells were serum starved for 24 h, treated with 25 ng/mL IL-33 for 24 h, harvested, and lysed. The levels of mRNA were determined using RT-PCR. (**D**) MCF7 cells were serum starved for 24 h, pretreated with the indicated concentrations of SP600125 for 2 h, and then exposed to 25 ng/mL IL-33 for 24 h, harvested, and lysed. The levels of mRNA were determined using RT-PCR. (**E**) Schema of the putative c-Jun-binding sites within the LPIN1 promoter region. (**F**) ChIP assay with either anti-c-Jun antibody or control mouse IgG, with input chromatin as a positive control. (**G**) Cells were serum starved for 24 h, pretreated with the indicated concentrations of SP600125 for 2 h, and then exposed to 25 ng/mL IL-33 for 24 h. Following that, the ChIP assay was performed on these samples using an anti-c-Jun antibody or control mouse IgG, with input chromatin as a positive control. The input DNA and DNA isolated from the precipitated chromatin were amplified using PCR and separated on a 1.5% agarose gel. IgG, immunoglobulin.

**Figure 5 cancers-13-02174-f005:**
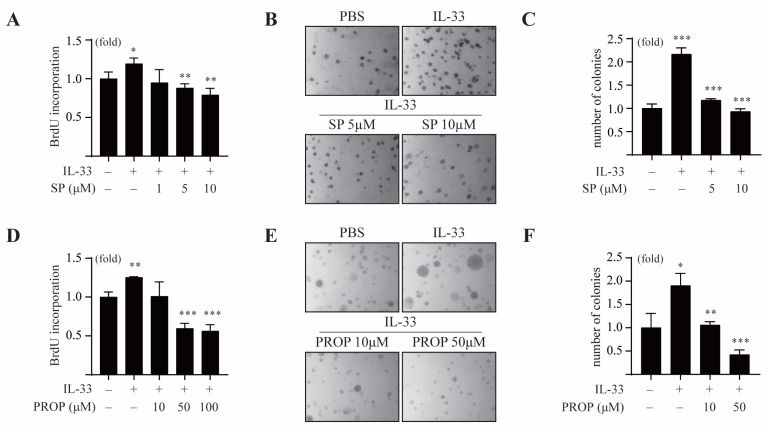
IL-33-induced LPIN1 promotes neoplastic cell transformation. (**A**) MCF7 cells were treated with different concentrations of SP600125 in the absence or presence of 25 ng/mL IL-33 for 48 h, following which the cell proliferation was estimated using the BrdU incorporation assay. (**B**,**C**) MCF7 cells were exposed to 25 ng/mL IL-33 with/without SP600125, as indicated in the soft agar matrix, and incubated at 37 °C in a 5% CO_2_-humidified atmosphere for 14 days. (**D**) MCF7 cells were treated with different concentrations of propranolol in the absence or presence of 25 ng/mL IL-33 for 48 h, following which the cell proliferation was estimated using a BrdU incorporation assay. (**E**,**F**) MCF7 cells were exposed to 25 ng/mL IL-33, with/without propranolol, as indicated in the soft agar matrix, and incubated at 37 °C in a 5% CO_2_-humidified atmosphere for 14 days. (**A**,**C**,**D**,**F**) Columns, mean of triplicate samples. bars, S.D. *, *p* < 0.05, **, *p* < 0.01, and ***, *p* < 0.001.

**Figure 6 cancers-13-02174-f006:**
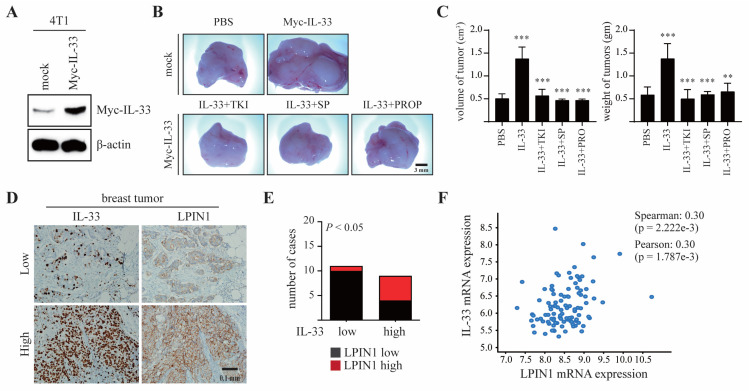
LPIN1 overexpression is associated with breast mammary tumorigenesis. (**A**) Expression levels of IL-33 in 4T1/mock and 4T1/Myc-IL-33. (**B**,**C**) Myc-IL-33-overexpressing 4T1 cells were treated with 100 μM TKI or 50 μM SP600125 or 0.5 mM propranolol. The treated cells were injected into the mammary glands of BALB/c mice and allowed to grow until tumors formed (14 days). Representative pictures of tumors (**B**) and tumor volume and weights (**C**) are shown. Columns, mean of triplicate samples. bars, S.D. **, *p* < 0.01, ***, *p* < 0.001. (**D**,**E**) Representative samples showing results of immunohistochemical analysis of breast-infiltrating duct carcinoma performed with the indicated antibodies on adjacent sections of the samples. In each sample, IL-33 and LPIN1 levels were semi-quantified in a double-blind manner as high or low, according to the standards presented and statistically analyzed in (**E**). Their correlation was analyzed using Fisher’s exact test (*p* < 0.05). (**F**) The correlation between IL-33 and LPIN1 mRNA expression was assessed using expression data from Molecular Taxonomy of Breast Cancer International Consortium (METABRIC) and TCGA (*n* = 6).

## Data Availability

The data presented in this study are available on request from the corresponding author.

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
