# Peer review of "IL-33-Induced Transcriptional Activation of LPIN1 Accelerates Breast Tumorigenesis"

_cancers, 2021, doi:10.3390/cancers13092174_

Round 1
Reviewer 1 Report
The authors describe that the tumorigenesis in breast cancer is guided by the IL-33-induced COT/JNK1/2 pathway in human breast cancer cell lines. The paper is well written. However, some parts (especially in vivo part) need to be clarified.
Minor revisions:
The experimental setting of in vivo experiments needed to be clarified in material and method sections, as well as in the results one. How long the cells were treated for? At what time the sacrifice of mice was accomplished? How many animals per conditions?
Figure 6. The legend is unclear. Panel E and F are not explained.
Reviewer 2 Report
The authors here present a paper about the role of IL33 in breast cancer tumorigenesis. Despite the role of IL33 and its receptor ST2, together with some of their downstream targets, have been already investigated in breast cancer progression by others and the authors themselves (PMID: 25531326), the work is well written and it shows a certain degree of novelty since it bridges the role of IL33 with LPIN1. In fact, the role of LPIN1 was already examined by the authors in the context of breast carcinomas (PMID: 27729374).
The experimental flow is complex and well-articulated, the overall approach is consistent and characterized by both in vitro and in vivo experiments.
On these bases, I suggest the editors to accept the paper in the present form for its further publication.
Reviewer 3 Report
The attached file shows the comments and suggestions.

Round 2
Reviewer 3 Report
The revised MS seems to be fine in this form.